# Mitophagy Regulates Neurodegenerative Diseases

**DOI:** 10.3390/cells10081876

**Published:** 2021-07-24

**Authors:** Xufeng Cen, Manke Zhang, Mengxin Zhou, Lingzhi Ye, Hongguang Xia

**Affiliations:** 1Department of Biochemistry & Research Center of Clinical Pharmacy of The First Affiliated Hospital, Zhejiang University School of Medicine, Hangzhou 310058, China; 0616502@zju.edu.cn (X.C.); 3150103967@zju.edu.cn (M.Z.); 0919897@zju.edu.cn (M.Z.); 0915580@zju.edu.cn (L.Y.); 2Liangzhu Laboratory, Zhejiang University Medical Center, 1369 West Wenyi Road, Hangzhou 311121, China

**Keywords:** mitophagy, neurodegenerative diseases, Parkinson’s disease, Alzheimer’s disease, Huntington’s disease, amyotrophic lateral sclerosis

## Abstract

Mitochondria play an essential role in supplying energy for the health and survival of neurons. Mitophagy is a metabolic process that removes dysfunctional or redundant mitochondria. This process preserves mitochondrial health. However, defective mitophagy triggers the accumulation of damaged mitochondria, causing major neurodegenerative disorders. This review introduces molecular mechanisms and signaling pathways behind mitophagy regulation. Furthermore, we focus on the recent advances in understanding the potential role of mitophagy in the pathogenesis of major neurodegenerative diseases (Parkinson’s, Alzheimer’s, Huntington’s, etc.) and aging. The findings will help identify the potential interventions of mitophagy regulation and treatment strategies of neurodegenerative diseases.

## 1. Introduction

Mitochondria are the primary source of cellular energy regulating cellular metabolism and physiology [1]. To maintain cellular metabolism and homeostasis, damaged or unwanted mitochondria should be eliminated through mitophagy, a form of mitochondrial quality control process [2]. Mitophagy is a highly selective autophagy process that eliminates dysfunctional or redundant mitochondria through multiple regulatory pathways in a ubiquitin-dependent or -independent manner [3]. Since the term “mitophagy” was first coined by Dr. Lemasters in 2005 [4], accumulating scientific evidence reveals that the accumulation of damaged mitochondria is one of the causal factors for various human diseases including neurodegenerative and cardiovascular diseases as well as cancers [5,6,7,8]. Among all the cell types affected by mitochondrial dysfunction, neurons are vulnerable to mitochondrial damage due to their high energy demand [9]. Therefore, it is necessary to understand the mechanisms of mitophagy in preserving mitochondrial health. Moreover, unraveling the role of mitophagy in the pathogenesis of neurodegenerative diseases is necessary. This paper reports the present knowledge of canonical and non-canonical mitophagy pathways. Specifically, we focus on how mitophagy-related genes contribute to different neurodegenerative diseases including Parkinson’s disease (PD), Alzheimer’s disease (AD), Huntington’s disease (HD), and amyotrophic lateral sclerosis (ALS).

## 2. Canonical and Non-Canonical Mitophagy

Accumulating evidence suggests that various signaling cascades regulate mitophagy. The canonical mitophagy, also called PINK/Parkin pathway-mediated mitophagy, promotes mitochondrial degradation in a ubiquitin-dependent manner and responds to cellular conditions damaging the mitochondria. Other mitophagy receptors are located on the mitochondrial outer or inner membrane; these mediate mitophagy in a ubiquitin-independent manner, which is classified as non-canonical mitophagy [10].

### 2.1. Ubiquitin-Dependent Pathways

#### 2.1.1. PINK1/Parkin-Dependent Pathway

Parkin (PARK2) is an E3 ubiquitin-protein ligase that has mutations implicated in the pathogenesis of Parkinson’s disease. PINK1 (phosphatase and tensin homologue (PTEN)-induced putative kinase 1) is a serine/threonine kinase monitoring mitochondrial integrity [11,12,13,14]. The canonical mitophagy pathway depends on the functions of these two proteins regulating the ubiquitin-dependent mitophagy pathway [15] (Figure 1a). PINK1 binds to the translocase of the outer membrane (TOM) complex on the outer mitochondrial membrane (OMM). It is transferred from the cytoplasm to the inner mitochondrial membrane (IMM), then cleaved by various proteases, transforming into a truncated form [16,17,18]. Eventually, it is released into the cytosol and degraded by the ubiquitin-proteasome system. This cellular process lowers the levels of PINK1 in healthy mitochondria [17]. When mitochondria are damaged, the membrane potential is depolarized, and PINK1 cannot be transported to the IMM but accumulates on the OMM. This accumulation leads to its autophosphorylation at Ser 228 and 402, which activates the PINK1 kinase. Subsequently, Parkin is recruited to the mitochondrial surface and initiates its E3 ubiquitin ligase activity by PINK1 mediated phosphorylation at Ser 65. Interestingly, Parkin with a mutation at Ser 65 can still be recruited since PINK1 phosphorylates ubiquitin at Ser 65, which also activates the E3 ligase activity of Parkin [19,20]. Activated PINK1 and Parkin catalyze the formation of phosphorylated poly-Ub chains connected by Lys 63 and Lys 27 to OMM proteins. Then, the autophagic receptors including P62/SQSTM1 and optineurin (OPTN) recognize phosphorylated poly-Ub chains, which directly bind to the autophagosomal light chain 3 (LC3) protein that anchors to the isolation membrane through the LIR motif to form autophagosome. The autophagosome fuses with the lysosome to initiate autolysosome, which degrades the damaged mitochondria [21,22,23]. 

Studies indicate that PINK1 and Parkin influence the morphology and dynamics of the mitochondria by regulating their fission/fusion. When the mitochondria are depolarized, PINK1 recruits Parkin, which ubiquitinates the mitochondrial fusion proteins (MFNs) to degradation. As a result, this inhibits the fusion of the damaged mitochondria, thereby degrading via the mitophagy pathway [24,25,26]. Additionally, PINK1 and Parkin limit damaged mitochondria to cellular-specific areas for mitophagy by suppressing mitochondrial transport. For instance, Miro, a component of the primary motor/adaptor complex transporting mitochondria through anchoring kinesin to the mitochondrial surface, is phosphorylated by PINK1 after mitochondrial damage, and then degraded through the Parkin-dependent ubiquitin-proteasome pathway [27,28]. Through inhibition of impaired mitochondria, mitophagy eliminates the damaged mitochondria caused by reactive oxygen species (ROS) [29]. In summary, to maintain energy balance and avoid oxidative stress, mammalian cells regulate mitochondrial network hemostasis via PINK1/Parkin pathway-mediated mitophagy, mitochondrial fission/fusion, and transportation.

#### 2.1.2. Parkin-Independent Pathways

Besides Parkin, several other E3 ligases are involved in the ubiquitin-dependent mitophagy pathways including Gp78, SIAH1, MUL1, and ARIH1 [5,30,31,32]. A recent study suggested that a mitochondrial Tu translation elongation factor (TUFm) interacts with PINK1 in a Parkin-independent manner to regulate mitophagy [33]. These proteins serve as an alternative part of Parkin in PINK1-mediated mitophagy, recruiting autophagic receptors including OPTN and NDP52. These receptors then recruit the autophagy factors including ULK1, DFCP1, and WIPI1 to focal spots proximal to the mitochondria for membrane elongation [34].

### 2.2. Ubiquitin-Independent Pathways

In contrast to the ubiquitin-dependent approach, a few mitophagy receptors directly mediate mitophagy via protein–protein interactions [10]. Several mitochondrial proteins comprise mitophagy receptors, targeting the damaged mitochondria into autophagosomes for degradation. These mitophagy receptor proteins including FUNDC1, NIX, BNIP3, etc., are located on the outer membrane of mitochondria; they interact with Atg8 family proteins through the LIR motif. Besides, they promote the engulfing of mitochondria by autophagosomes, thereby causing mitophagy [35]. 

#### 2.2.1. Ubiquitin-Independent Mitophagy in *Saccharomyces cerevisiae*

Previous autophagy studies have been primarily conducted in a yeast cell system. So far, more than 30 autophagy-related (Atg) proteins have been identified in yeast [35]. Atg32 regulates mitophagy in yeast cells [36,37]. Notably, Atg32 is a mitochondrial outer membrane protein with its *N*-terminal located in the cytoplasm, comprising a WXXL-like Atg8-binding motif, suggesting its interaction with Atg8 family proteins [37,38]. In the post-log growth phase of yeast cells, Atg32 participates in mitophagy by degrading a portion of the mitochondria [37]. Despite Atg32 participating in mitophagy regulation, its contribution does not affect the level of mitophagy induced by nutrient deprivation [37]. After the initiation of mitophagy, Atg32 is imported into the autophagosomes together with mitochondria by binding to Atg11, an adaptor protein of selective autophagy [36] (Figure 1b).

#### 2.2.2. Ubiquitin-Independent Mitophagy in Mammalian Cells

In mammalian cells, the OMM proteins including NIX (NIP3-like protein X, also known as BNIP3L), BNIP3 (Bcl-2 and adenovirus E1B 19 kDa-interacting protein 3), and FUNDC1 (FUN14 domain containing 1) are identified as ubiquitin-independent mitophagy receptors. These mediate mitophagy in response to various conditions including hypoxia, stress, and cell differentiation (Figure 1b). 

NIX and BNIP3, both related to the Bcl2-family protein, share 56% homology in amino acid sequence [39]. Studies show that NIX binds to LC3/GABARAP via its LIR motif located at its *N*-terminal, thereby causing autophagosomal membrane formation [40]. Additionally, genetic studies confirm that NIX deficiency triggers mitochondrial accumulation in mature erythrocytes, inducing anemia in mice [41,42]. As a mitophagy receptor, BNIP3 also contains an LIR motif and functions similarly to NIX. Interestingly, BNIP3–LC3 interaction induces mitophagy and the clearance of unwanted endoplasmic reticulum (ER). On the other hand, BNIP3 also plays a role in apoptosis, indicating that mitophagy and apoptosis harbors a crosstalk regulation [43,44]. Moreover, under hypoxic conditions, both NIX and BNIP3 mediated-programmed mitophagy are implicated in cellular respiration and differentiation [45,46]. Furthermore, a recent study revealed that BNIP3-mediated mitophagy plays a protective role via inhibition of apoptosis and ROS production in an ischemia/reperfusion (I/R)-induced acute kidney injury model [47]. Besides these two proteins, accumulating studies suggest additional Bcl2-family proteins participating in mitophagy regulation. Bcl2, also known as an anti-apoptosis protein, negatively regulates mitophagy by inhibiting the Parkin function via direct protein–protein interaction [48]. MCL-1 is another vital anti-apoptosis protein, identified recently as a novel mitophagy receptor. This protein triggers mitophagy in an Alzheimer’s disease mouse model through the action of UMI-77 [49].

FUNDC1 is another OMM protein identified as a receptor for hypoxia-induced mitophagy [50]. FUNDC1 locates only on the mitochondrial membrane, with its *N*-terminal exposed to the cytosol and *C*-terminal protruding into the intermembrane space [50]. Similar to other mitophagy receptors, FUNDC1 contains a typical LC3 binding motif (LIR) interacting with Atg8 family proteins and subsequently inducing mitophagy [50]. Mechanistic studies reveal that FUNDC1 phosphorylation at the Tyr18 by Src kinase inhibits mitochondrial clearance, whereas FUNDC1 dephosphorylation by PGAM5 promotes mitophagy [51,52]. Additionally, a growing body of evidence suggests that FUNDC1 participates in mitochondrial dynamics by interacting with mitochondrial fusion/fission-related proteins OPA1 (optic atrophy 1) and Drp1 (dynamin-related protein 1) [53,54]. Furthermore, physiological function investigations show that FUNDC1-mediated mitophagy ameliorates myocardial reperfusion and kidney injuries [55,56]. 

AMP-activated protein kinase (AMPK) is a vital energy regulatory factor, and it is a heterotrimer complex-formed by three different subunits: α-subunit, β-subunit, and γ-subunit [57]. α-subunit contains the kinase domain, the β-subunit contains a carbohydrate-binding module [58], while the γ-subunit is composed of four (tandem cystathionine-β-synthase) CBS domains, which can bind adenine nucleotides to sense the ratio of ATP and AMP [59]. Once AMPK is activated, it phosphorylates downstream substrates, thus reconstituting metabolism. AMPK is involved in the regulation of various metabolic processes such as glucose metabolism, lipid metabolism, autophagy, and mitophagy. AMPK also participates in the regulation of mitochondrial quality control including mitochondrial biogenesis, mitochondrion fission, and mitophagy. Exercise can promote mitochondrial biogenesis by activating AMPK [60]. The overexpression of the constitutively active AMPK γ3-subunit in mice can also promote mitochondrial biogenesis [59]. Mechanistic studies have revealed that AMPK regulates mitochondrial biogenesis by regulating PGC-1α through direct phosphorylation or indirect pathway [59,61]. In addition, AMPK may promote mitochondrion fission by phosphorylating Ser 155 and Ser 173 of the mitochondrial fission factor (MFF) [62,63] and promotes the recruitment of dynamin-related protein 1 (DRP1) to mitochondria. Moreover, AMPK regulates autophagy via reducing mTOR activity by phosphorylating the mTORC1 subunit RAPTOR and the mTOR upstream regulator TSC2 [64,65]. On the other hand, AMPK promotes the degradation of damaged mitochondria by phosphorylating ULK1 [60]. Taken together, AMPK promotes the renewal of mitochondria by regulating the mitochondrial quality control system from biogenesis to mitophagy.

On the other hand, recent studies have found that phosphatase and tensin homolog (PTEN)-long (PTEN-L) plays an important role in the negative regulation of mitophagy. PTEN-L is an isoform of PTEN with the addition of 173 to its *N*-terminus. PTEN-L localizes at the outer mitochondrial membrane (OMM) and reduces parkin E3 ligase activity by decreasing parkin phosphorylation. In addition, PTEN-L reduces the level of phosphorylated ubiquitin (p-Ser65-Ub). Through these effects, PTEN-L inhibits mitophagy. This discovery provides new insights into the regulation of mitophagy [66].

## 3. Mitophagy in Aging and Neurodegenerative Diseases

### 3.1. Aging

Mitochondria play an essential role in the process of aging. The function of mitochondria decreases, and mitochondrial DNA mutations accumulate with aging. In the process of aging, the feature of dysfunctional mitochondria includes decreasing the content of mitochondria, changing the morphology of mitochondria, reducing the efficiency of the electron transport chain, and increasing ROS production. The mitochondrial quality control system, especially mitophagy, decreases with the deepening of aging. These changes are accompanied by the occurrence and development of diseases. Due to the high energy consumption of neurons, mitochondria are particularly important for their function, indicating that aging is a significant risk factor for neurodegenerative diseases. 

In recent studies, the induction of mitophagy and degeneration of NAD^+^ in Werner syndrome (WS) patients, which is an autosomal recessive accelerated aging disease and caused by mutations in the gene encoding the Werner (WRN) DNA helicase, has been observed [67]. The main clinical symptoms include cancer, juvenile cataracts, dyslipidemia, premature atherosclerosis, and insulin resistance diabetes. NAD^+^ supplementation can significantly relieve the accelerated aging process in Caenorhabditis elegans and Drosophila melanogaster models of WS [67]. Through the mechanism study, the NAD^+^ effect is achieved through DCT-1 and ULK-1 dependent mitophagy [67]. Cardiovascular aging is another very important aging event in which the regulation of mitochondrial homeostasis is involved [68]. Many studies have shown that mitophagy plays an important role in the anti-cardiovascular aging process in recent years. Heat shock protein 27 (HSP27), a small heat shock protein involved in the responses to oxidative stress, heat shock, and hypoxic/ischemia injury [69], can induce mitophagy and antioxidant function to reduce the degree of heart aging [70]. Another study demonstrated that double knockout of Akt2 and AMPK induced cardiac aging in 12-month-old mice, which was most likely achieved by reducing autophagy and mitophagy levels because more p62, lower LC3B II and LC3B I ratios, and lower level of mitophagy receptors associated with aging including PINK1, Fundc1, etc. were observed in double knockout Akt2 and AMPK mice [71]. Harman’s free radical theory is the commonly accepted aging theory at present [72,73]. This theory assumes that the decrease in cellular longevity is caused by the increase in reactive oxygen species. Recent studies have shown that mitochondria are the main source of ROS and the main target of ROS-mediated damage [74]. ROS is a by-product of mitochondrial respiration. With aging, mitochondria function decreases, and more ROS are accumulated, leading to the damage of mitochondria and mtDNA. These phenomena suggest that increasing mitophagy and restoring mitochondrial function can prevent and treat vascular and cardiac aging-related dysfunction.

Although growing studies have shown that there is a strong relationship between mitophagy and aging, the reason for the decline in mitophagy with aging is not clear. With aging, ROS accumulation may lead to the oxidation of many proteins related to mitophagy function including PINK1, Parkin, LC3, etc. The oxidation of protein will reduce its function, which may lead to mitophagy dysfunction. S-nitrosylation is a critical post-translational regulation of most proteins, which attaches a nitrogen monoxide group to the thiol side chain of cysteine [75]. Recent studies have shown that it may play an important role in aging and neurodegenerative diseases [76]. S-nitrosoglutathione reductase (GSNOR) is an important enzyme regulating S-nitrosylation. Its activity gradually decreases with aging, and leads to mitophagy related protein such as Drp1 and Parkin S-nitrosylation, thus affecting the function of mitophagy [77]. Another study on Alzheimer’s disease (AD) revealed that the S-nitrosylation transfer reaction mediated by UCH-L1, Cdk5, and Drp1 may play an important role in the occurrence and development of AD [76]. These results indicate that with the development of aging, post-translational modification may play a critical role in the dysfunction of proteins, which affects the physiological process, especially mitophagy.

In addition, skeletal muscle aging is also related to mitophagy. Sarcopenia refers to the loss of muscle mass and function with aging, and its molecular mechanism is not clear. However, an increasing body of evidence has shown that it is related to decreased autophagy levels, especially mitophagy. Mitochondrial dysfunction and fragmentation exist in aging muscles, which indicates that the regulation of mitochondrial homeostasis is unbalanced. Some studies have reported that NIX, PINK1, and Parkin levels are enhanced in aging muscles, suggesting that mitophagy as an anti-aging process is activated with aging [78,79,80]. Although mitophagy can remove damaged mitochondria and restore mitochondria health, with the deepening of the aging process, the related functional proteins of mitophagy are oxidized, resulting in the weakening of function, which is not enough to clear the damaged mitochondria, thus accelerating the aging. Exercise is thought to be a way to enhance mitophagy. Studies have shown that Parkin plays an important role in exercise-induced mitophagy [79], suggesting that Parkin, as a regulatory protein of mitophagy, participates in the process of mitophagy in skeletal muscle and plays an critical role in the aging of skeletal muscle. These studies indicate that the induction of mitophagy has a significant effect on the treatment of aging-related diseases. A clinical trial on Urolithin A (UA) shows that UA, as a safe and effective inducer of mitophagy, plays an important role in skeletal muscle health, providing more evidence for the treatment of premature aging-related disease by the induction of mitophagy [81].

### 3.2. Parkinson’s Disease

Parkinson’s disease (PD) is a prevalent neurodegenerative disorder primarily characterized by loss of dopaminergic neurons in the substantia nigra and accumulation of mutational alpha-synuclein. This was first described by James Parkinson in 1917 [82]. The precise mechanism of PD is unclear, but considerable evidence suggests that damage in mtDNA, redundant ROS, and dysfunctional mitophagy potentially regulates the occurrence of PD [83,84]. Accumulation of mitochondrial DNA (mtDNA) mutations caused by reactive oxygen species (ROS) results in mitochondrial dysfunction, thereby enhancing ROS production [85]. Mitophagy, a crucial mitochondrial quality control process, regulates mitochondrial function for neuron health [86]. Accumulation of damaged and dysfunctional mitochondria has been observed in Parkinson’s disease, suggesting that mitochondrial network homeostasis is impaired in PD patients [87,88]. Furthermore, PINK1 knockout mice revealed a progressive loss of dopaminergic neurons in the substantia nigra [89]. This suggests that the disorder of the mitophagy process is potentially and strongly associated with PD (Table 1).

PINK1-Parkin pathway mutations inhibit mitophagy, which is directly related to PD occurrence [38]. PINK1 is highly expressed in organs or tissues with high energy demand including the brain, heart, and muscles. Moreover, Parkin is expressed in various types of tissues, which perhaps shows its complex functions [103]. Parkin mutations related to PD prevent the recruitment of Parkin to mitochondria and the accumulation of damaged mitochondria. This enhances ROS production, thereby promoting PD pathologies [92]. Moreover, mitochondrial disturbance of fission and fusion caused by alpha-synuclein can be rescued via PINK1 and Parkin co-expression [104]. Additionally, the NIX-mediated mitophagy pathway independently restores mitophagy in the PD patient cell lines without functional PINK1 and Parkin [105]. On the other hand, USP30 is identified as a deubiquitinase for mitophagy regulation negatively. Overexpression of USP30 inhibits mitophagy by removing ubiquitin on damaged mitochondria [106]. Several USP30 inhibitors are under development for the treatment of PD [107]. Despite PINK1/Parkin pathway dysfunction being a major contribution to PD pathologies, more studies have shown other genes that influence mitophagy involved in PD. DJ-1 is a mitochondrial location redox sensor. Loss of DJ-1 leads to mitochondrial fragmentation that may affect mitophagy. Mutation of DJ-1 causes a recessive form of PD [108,109]. Mutation of LRRK2, a large multidomain protein, influences mitophagy via regulating the PINK1/Parkin pathway, causing an autosomal dominant form of PD [110,111,112]. These findings suggest that mitochondrial dysfunction is strongly related to PD pathogenesis, and induction of mitophagy rescuing mitochondrial biogenesis may ameliorate PD pathology. 

Notably, most PD patients are classified as sporadic patients, whereas only less than 10% of PD cases are diagnosed as familial PD. Among familial PD patients, important mutations in DJ-1 and GBA are implicated in maintaining normal mitochondrial function [94,95]. Although the biogenesis of these two categories is different, a significant difference between both groups for clinical profile or motor symptoms cannot be observed [113]. Since the pathogenic cause of PD is complicated and is still unknown, an effective strategy that can radically cure PD remains unavailable [114]. By eliminating dysfunctional mitochondria and degrading abnormal structural proteins, mitophagy is a potential strategy for PD treatment.

### 3.3. Alzheimer’s Disease

Alzheimer’s disease (AD) is the most common neurodegenerative disease; its symptoms include memory loss and cognitive impairments. Dysfunctional mitochondria accumulation, damaged synapse, disease-defining amyloid-β (Aβ) oligomers, and intracellular neurofibrillary tangles (NFTs) are the fundamental pathological hallmarks of AD [9,115].

A series of evidence suggests that amyloid deposition is a common pathological hallmark in numerous neurodegenerative diseases including AD; an excessive aggregation of amyloid-beta impairs neurons, causing their death [116]. Besides, amyloid precursor protein-derived *C*-terminal fragments (APP-CTFs) accumulating in AD patients and AD mouse models trigger mitochondrial damage and mitophagy failure in an Aβ-independent manner [117]. Increasing evidence shows the existence of a strong relationship between mitophagy failure and neuron degeneration. Studies by Fang et al. showed that mitophagy is reduced in APP/PS1 mouse model, Aβ-based *C. elegans* model, and even in the hippocampus of AD patients’ brains [118]. Another study also showed mitochondria fragmentation and dysfunction in Aβ expression of *Drosophila melanogaster*, suggesting that mitophagy failure may be a hallmark of AD [119]. Considering that mitochondria regulate energy generation in neurons, dysfunctional mitochondria will badly influence the signal delivery from one neuron to another. In addition, hyper-phosphorylated Tau, as another hallmark protein in AD, seemingly aggregates in AD patients and the loss of Tau is neuroprotective [120]. Numerous findings suggest that Tau expression leads to impairment in mitophagy [121,122,123]. This indicates that protein aggregate degradation is a potential strategy to reduce their impairment to CNS and protect neurons. 

On the other hand, mitochondria are highly dynamic organelles; their shape and size, distributive situation, and physiological functions are regulated by their fission and fusion [124]. These processes include Drp1-mediated fission and OPA1-mediated fusion of mitochondria. Recent studies have reported that both levels of Drp1 and OPA1 are remarkably decreased in AD [125,126]. This indicates that the imbalance between fission and fusion affects the normal structure and function of mitochondria, promoting AD pathology. Mitophagy and mitochondrial dynamics interact with each other to maintain a healthy mitochondrial recycling balance [127]. Based on accumulating evidence, abnormal structures, functional defects, and variations in mitochondrial dynamics, and decline in the level of mitophagy are observed in neurons of AD patients [128,129]. Moreover, as terminally differentiated cells, neuronal cells are susceptive to various types of mitochondrial dysfunctions and irreversible damage, eventually leading to neuron death once compromising mitophagy cannot recycle damaged or redundant mitochondria properly. 

Recent research has reported compounds including beta-Asarone and UMI-77 that help improve the learning and memory of AD mice as well as ameliorate disease pathologies by promoting mitophagy [49,130]. Another two compounds (nicotinamide riboside and urolithin A) are also reported to induce mitophagy. Nicotinamide riboside is a precursor of NAD^+^ and can be metabolized to produce NAD^+^ in cells and reduces Aβ levels in APP/PS1 mice [131,132,133]. This suggests that enhancement of NAD^+^ may be beneficial for AD treatment. Urolithin A (UA) is a natural compound that ameliorates cognitive decline in the APP/PS1 mouse model via mitophagy activation [81,118,134]. A recent study showed that rapamycin, an mTOR inhibitor that can induce autophagy, also induces mitophagy and alleviates cognition in a mouse model of Alzheimer’s disease [135]. This suggests that mitochondrial dysfunction is the most prominent feature of AD, whereas induction of mitophagy appears as a potential strategy for AD treatment.

### 3.4. Huntington’s Disease

Huntington’s disease (HD) is a rare autosomal dominant disorder caused by an expansion of cytosine-adenine-guanine (CAG) repeats within the huntingtin (Htt) gene. This results in polyglutamine (polyQ) expansion in the encoded huntingtin protein. Since the clinical syndrome of HD displays apparent neuropathic traits including motor dysfunction, cognitive decline, and psychiatric disturbances, it can also be classified into neurodegenerative diseases. The prevalence of Huntington’s disease is estimated at 4–10 per 100,000 in the Western world and the onset time and severity of HD is positively related to the length of CAG repeats [136]. Although the pathogenesis of HD remains unclear and lacks effective therapeutic methods, increasing evidence reveals that mitochondria regulate the HD pathology process.

Aberrant mitochondrial morphology, fragmentation, and decreased mitochondrial mass are observed in HD patients. Besides, the mutant huntingtin severely impairs mitochondrial respiration and ATP production, suggesting that energy metabolism in HD may fall into disorder [137]. As for the fragmented mitochondria in HD pathology, scientists believe that excessive mitochondrial fission is potentially caused by increasing levels of Drp1 and decreasing levels of OPA1 and mitofusin 1 (Mfn1) [98,138,139]. These findings show that mutant huntingtin impairs mitochondria by disturbing mitochondrial dynamics, further influencing its function. This indicates that functional mitochondria restoration might be an effective treatment for HD. In line with the findings by Khalil’s [140] group, PINK1 overexpression, which regulates Parkin-mediated mitophagy, partially restored mitophagy and promoted neuroprotection in Huntington’s disease. Nonetheless, Guo et al. [99] suggested that accumulation of valosin-containing protein (VCP), an mtHtt-binding protein on the mitochondria, induces superabundant mitophagy, causing the death of neurons. Moreover, Rhes, a type of GTPase, was reported to upregulate mitophagy via the NIX receptor. This led to striatal cell death and striatal lesions, speculating that exaggerated mitophagy might be a contributing factor of HD [100,141]. Overall, abnormal mitochondrial size and morphology have been confirmed in HD, but the role of mitophagy (i.e., eliminating dysfunctional and unwanted mitochondria) remains controversial. 

### 3.5. Amyotrophic Lateral Sclerosis

Amyotrophic lateral sclerosis (ALS, also known as motor neuron disease) is a progressive and ultimately fatal, age-dependent neurodegenerative disorder with motor neuron degeneration. Its prevalence stands at 6–10 per 100,000. Out of all ALS cases, a majority (90%) are classified as sporadic ALS (SALS), while only 10% are diagnosed as familial ALS (FALS) [142]. The precise mechanism of ALS remains unclear, but most ALS related genes are implicated in mitophagy regulation including superoxide dismutase 1 (SOD1) gene [143], TAR DNA binding protein 43(TDP-43) gene [144], fused in sarcoma/translated in liposarcoma (FUS/TLS) gene [145], optineurin (OPTN) gene [101], and TANK-binding kinase 1 (TBK1) gene [146], etc. Notably, OPTN mediates the formation of autophagosome that devours the unwanted mitochondria, then sequesters it from the cytosol, ensuring subsequent degradation [147]. Three types of OPTN mutations have been reported in ALS patients [101], strongly suggesting that OPTN is related to the etiology of ALS by affecting mitophagy [148]. TBK1 is another key protein that potentially promotes ALS pathology by affecting mitophagy. TBK1 activates mitophagy by phosphorylating and recruiting OPTN to depolarized mitochondria [149]. Besides, TBK1 phosphorylates recruit other receptors including NDP52, TAX1BP1, and p62 to mitochondria by ubiquitin-binding and mediates engulfment through their relationship with LC3 [150]. 

On the other hand, as noted in PD, AD, and HD, the accumulation of neurotoxic misfolded proteins and aggregates within motor neurons is a primary pathological hallmark of ALS [151]. SOD1 and TDP-43-associated aggregation are the vital protein aggregates in ALS patients. Given that mitophagy may also promote the elimination of protein aggregates, mitophagy induction might ameliorate ALS pathology. 

## 4. Discussion

Due to the indispensable role of mitochondria in energy generation and metabolic regulation, increasing attention has been drawn to mitophagy regulation in the maintenance of cellular homeostasis and its function under pathological conditions. Nevertheless, an integral regulatory system of mitophagy and the crosstalk between known mitophagy pathways remain unreported. In this review, we summarize the ubiquitin-dependent and -independent pathways implicated in the regulation of mitochondria removal; these might help understand the molecular mechanism of mitophagy. 

Neurodegenerative disorders are a diverse group of diseases including Parkinson’s disease, Alzheimer’s disease, Huntington’s disease, and amyotrophic lateral sclerosis, etc. These are characterized by different aspects of abnormal mitochondria, ranging from morphology (shape and size) to number (too much or too few), mitochondrial dynamics (imbalance between fission and fusion), function (short of energy supply), and mitophagy (deficient or excessive). On the other hand, specific misfolded and abnormally accumulating proteins have also been noted in these diseases. For example, alpha-synuclein in Parkinson’s disease; amyloid-beta (Aβ) and hyperphosphorylated tau (p-tau) in Alzheimer’s disease; mutant huntingtin proteins in Huntington’s disease, and mutant superoxide dismutase 1 (SOD1) and TAR DNA binding protein 43 (TDP-43) in amyotrophic lateral sclerosis. Mitophagy mediated by the PINK1/Parkin pathway plays an important role in the mitochondria quality control system and in eliminating misfolded and unwanted proteins deposited in mitochondria. Although the exact mechanisms of these diseases remain unclear, the induction of mitophagy, as a potential approach to maintaining a healthy mitochondrial and proteinic level, is a promising strategy for symptomatic treatment. According to recent research, compounds regulating mitophagy have been confirmed to help improve the pathologies of these diseases in mice models. Additional studies are necessary to elucidate the mechanisms of mitophagy and clarify the precise role of mitophagy in neurodegenerative diseases. Furthermore, since abnormal energy metabolism is a common phenomenon in many other types of diseases including cancers and cardiovascular diseases, induction or inhibition of mitophagy is a potential method for restoring energy metabolism homeostasis for treatment.

## Figures and Tables

**Figure 1 cells-10-01876-f001:**
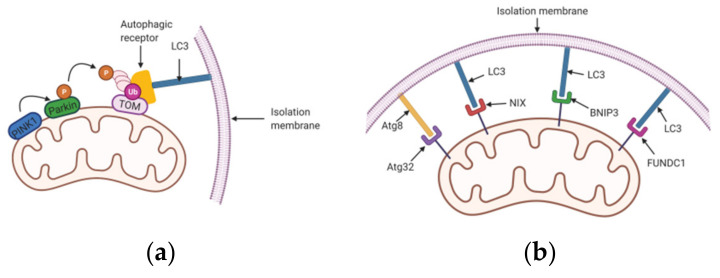
The PINK1/Parkin and ubiquitin-independent pathways of mitophagy. This figure was created with BioRender.com. (**a**) PINK1 accumulates on the damaged mitochondria and recruits E3 ligase Parkin. Autophagic cargo receptors recognize phosphorylated poly-Ub chains catalyzed by activated PINK1 and Parkin, thereby facilitating the mitochondria into autophagosomes fusing with a lysosome, causing degradation of mitochondria. (**b**) The ubiquitin-independent pathways are mediated by mitophagy receptors Atg32, NIX, BNIP3, and FUNDC1, etc. In yeast cells, OMM protein Atg32 binds to Atg8 on the depolarized mitochondrial membrane via its WXXL-like Atg8-binding motif and eliminates unwanted mitochondria. In mammalian cells, mitophagy receptors NIX, BNIP3, and FUNDC1 directly interact with LC3 through the LIR motif to mediate mitophagy.

**Table 1 cells-10-01876-t001:** Genes related to neurodegenerative diseases and mitophagy.

Gene	Protein	Function in Mitophagy	Disease	Reference
*PARK6*	PINK1	Kinase, involved in the regulation of several mitophagy related proteins	PD, AD, HD	[90,91]
*PARK2*	Parkin	Selectively recognize and eliminate damaged mitochondria from the cell	PD, AD, HD	[91,92]
*SNCA*	Alpha-synuclein	Located on the mitochondria through its *N*-terminal, lead to mitochondrial damage and dysfunction	PD	[93]
*DJ-1*	Protein DJ-1	Regulate mitophagy and ATP produce	PD	[94]
*GBA*	Glucocerebrosidase	Ensure normal function of lysosome and influence mitochondrial morphology and dynamics	PD	[93,95]
*DRP1*	Dynamin-related protein 1	Mediate mitochondrial fission	PD, AD, HD	[96]
*OPA1*	Optic atrophy 1	Mediate mitochondrial fusion	PD, AD, HD	[97]
*MFN1*	Mitofusin 1	Mediate mitochondrial fusion	PD, AD, HD	[98]
*VCP*	Valosin-containing protein	Accumulation of VCP can induce superabundant mitophagy	HD	[99]
*Rhes*	Ras homolog enriched in striatum	Up-regulate mitophagy via NIX receptor	HD	[100]
*OPTN*	Optineurin	Mediates the formation of autophagosome	ALS	[101]
*TBK1*	TANK-binding kinase 1	Mediate the engulfment of damaged mitochondria	ALS	[102]

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
