# Peer review of "Mitophagy Regulates Neurodegenerative Diseases"

_cells, 2021, doi:10.3390/cells10081876_

Round 1

Reviewer 1 Report

This is a comprehensive, descriptive review article on mitophagy mechanisms and their potential involvement in neurodegenerative diseases. The article's flow is adequate and the article is comprehensive. However, it does not represent anything new in the field and summarizes what has already been written. There are several review articles already published on this topic, and I don't see the need for another one. However, I recognize that this is an Editorial decision, as the Editors have commissioned a special volume devoted to mitophagy. In that context, this review article is meaningful.

Author Response

Thank you for your comments

Reviewer 2 Report

The authors Xufeng Cen et al in their manuscript entitled ‘Mitophagy regulates neurodegenerative diseases’ have extensively documented the recent advances in understanding the potential role of mitophagy in the pathogenesis of major neurodegenerative diseases. The review is an in-depth description of canonical and non-canonical forms of mitophagy. The authors further discuss how the defect in mitophagy is a major contributing factor in multiple neurodegenerative diseases (Parkinson’s disease, Alzheimer’s disease, Huntington’s disease, and Amyotrophic lateral sclerosis). The article is exhaustive and discusses the implications of mitophagy in neurodegeneration in great detail. The review includes good illustrations at appropriate places which make the respective points very clear.  

Author Response

Thank you for your recognition of our article

Reviewer 3 Report

This manuscript doesn't contain any updated knowledge or information and very similar to previous review paper titled "Mechanisms and roles of mitophagy in neurodegenerative diseases."

Author Response

Thank you for your comments. We have added some new contents: AMPK and PTEN-L regulate mitophagy, which are marked in red.

Reviewer 4 Report

The authors have written a concise review on mitophagy focussing on beurodegenerqtive diseases. 
I have some suggestions to enhance the content.
The authors did not mention on negative regulators of mitophagy such as PTENL. It is good to discuss on the negative regulators in a separate section
AMPK mediated mitophagy has gained lot of attention in recent years. The authors should include sepearate section on AMPK axis.

Author Response

Thank you for your suggestion. We have added AMPK and PTEN-L contents according to your suggestion and marked them in red. Since the regulation of mitophagy by these two protein is not the focus of our review, we do not have separate section on AMPK axis.

Reviewer 5 Report

In this manuscript, Xufeng Cen et al. have reviewed the process of mitophagy in cases of neurodegenerative disorders. The article describes some signaling pathways involved in mitophagy regulation and the pathogenesis of neurodegenerative diseases such as PD, AD, HD and ALS.

The article is of broad interest and well written. In my opinion, there is one important topic that has been omitted and cannot be ignored. Mitophagy is not a phenomenon exclusively related to pathologies, but a normal process during the lifetime, to the point that mitochondrial dysfunction is considered a hallmark of aging. This matter should be addressed as a subtitle before the above-mentioned neurodegenerative diseases and could include premature aging diseases as an introductory section to the main topic of the review.

Author Response

Thank you for your suggestion. Given the complex and comprehensive relationship between mitochondria and aging, which is not the focus of our review, we only added a brief introduction before the neurodegenerative diseases section. We also have added some new contents: AMPK and PTEN-L regulate mitophagy, which is marked in red. We hope that this addresses the reviewer’s concern.

Round 2

Reviewer 3 Report

The authors manuscript is very similar to the published manuscript "Wang Y, Liu N, Lu B. Mechanisms and roles of mitophagy in neurodegenerative diseases. CNS Neurosci Ther. 2019 Jul;25(7):859-875. doi: 10.1111/cns.13140. Epub 2019 May 2. PMID: 31050206; PMCID: PMC6566062." Moreover, in their 2nd revised manuscript the authors have added only 38 lines. It is not enough to publish a new updated review paper.

Author Response

Thank you for your comments. According to your suggestions, I have expanded  my review and added “Mitophagy in aging”section. Please refer to the revised manuscript for details.